# Coronary Stent Strut Fractures: Classification, Prevalence and Clinical Associations

**DOI:** 10.3390/jcm10081765

**Published:** 2021-04-19

**Authors:** Katharina Schochlow, Melissa Weissner, Florian Blachutzik, Niklas F. Boeder, Monique Tröbs, Liv Lorenz, Jouke Dijkstra, Thomas Münzel, Stephan Achenbach, Holger Nef, Tommaso Gori

**Affiliations:** 1Kardiologie 1, Zentrum für Kardiologie, Universitätsmedizin der Johannes Gutenberg-Universität Mainz, Langenbeckstraße 1, 55131 Mainz, Germany; katha@web.de (K.S.); mweissner@web.de (M.W.); livlorenz@web.de (L.L.); tmuenzel@unimedizin-mainz.de (T.M.); 2Med. Klinik 2, Universitätsklinikum Erlangen, Ulmenweg 18, 91054 Erlangen, Germany; fblachutzik@web.de (F.B.); mtroeb@web.de (M.T.); stephanachenbach@web.de (S.A.); 3Med. Klinik I, Universitätsklinikum Gießen und Marburg GmbH, Klinikstr. 33, 35392 Gießen, Germany; nilasboeder@web.de (N.F.B.); holgernef@web.de (H.N.); 4Division of Image Processing, Department of Radiology, Leiden University Medical Center, 2300 RC Leiden, The Netherlands; jdijkstra@hotmail.com

**Keywords:** coronary stent, bioresorbable scaffold, optical coherence tomography, stent thrombosis, stent restenosis

## Abstract

Introduction. The frequency, characteristics and clinical implications of Strut fractures (SFs) remain incompletely understood. Methods and results. A total of 185 (160 patients) newer-generation drug-eluting stents (DES) were imaged. SFs were found in 21 DES (11.4%) and were classified in four patterns: one single stacked strut (41%); two or more stacked struts (23%); deformation without gap (27%); transection (9%). In multivariable analysis, calcific and bifurcation lesions were associated with SF in DES (OR: 3.5 [1.1–11] and 4.0 [2.2–7.2], *p* < 0.05). Device eccentricity and asymmetry as well as optical coherence tomography (OCT) features of impaired strut healing were also associated with SF. The prevalence of fractures was similar in a set of 289 bioresorbable scaffolds (BRS). In a separate series of 20 device thromboses and 36 device restenoses, the prevalence of SF was higher (61.2% of DES and 66.7% of BRS, *p* < 0.001 for both), with a higher frequency of complex SF patterns (*p* < 0.0001). In logistic regression analysis, fractures were a correlate of device complications (*p* < 0.0001, OR = 24.9 [5.6–111] for DES and OR = 6.0 [1.8–20] for BRS). Discussion. The prevalence of OCT-diagnosed SF was unexpectedly high in the setting of elective controls and it increased by about three-fold in the setting of device failure. Fractures were associated with increased lesion complexity and device asymmetry/eccentricity and were more frequent in the setting of device failure such as restenosis and thrombosis.

## 1. Introduction

Stent-related adverse events, including those that occur late after implantation, remain an issue in interventional cardiology. In a recent individual-patient data-pooled study analysis, very-late stent-related events had an incidence of ∼2%/year with all stent types, which did not appear to decrease over time. The mechanism of these events is complex and multifactorial [1]. Stent strut fracture (SF) of drug-eluting stents (DES) stimulates neointima hyperplasia [2] and is one of the proposed mechanisms of in-stent restenosis, stent thrombosis and target lesion/vessel failure [3,4,5,6,7,8,9,10,11,12,13]. In cross-sectional studies, SFs appear to be associated with evidence of inflammation/hypersensitivity and with positive remodeling as well as evaginations, all of which are acknowledged correlates of late stent thrombosis [14]. Because of the low sensitivity of angiography, the incidence, predictors, and implications of SF remain incompletely explored. In the literature, the incidence of SF varies from <1% to 20% depending on the time after implantation, stent type and, most importantly, the methods and definitions used for the diagnosis of SF. In the largest available series (6555 patients and 16,482 stents), SF were found in 12% of the patients and 22% of the stents [15] and were associated with a more than three-fold increase in the incidence of restenosis, target lesion revascularization, and stent thrombosis. In analogy, SFs were found in as many as 29% of drug-eluting stents in an autopsy series. In this paper, the authors classify SFs in grades and show that as many as 67% of the stents presenting an SF were associated with pathologic evidence of restenosis or thrombosis at the fracture sites [16].

Optical coherence tomography (OCT) allows the in vivo diagnosis of SF with a resolution that is superior to high-contrast radiography used in ex vivo samples. We set out to provide a systematic analysis of the prevalence, patterns, possible predictors, and clinical correlates of SF.

Further, since SFs represent a step of the programmed resorption of bioresorbable scaffolds (BRS), these devices (although not anymore on the market) provide an ideal bench to investigate the clinical implications of SF in device failure.

## 2. Materials and Methods

### 2.1. Objective of the Study

We set out to describe the prevalence of SF, classified in 4 different patterns of increasing severity, as an incidental finding in the setting of elective planned controls. We investigated the associations of SF with clinical, procedural, and OCT parameters. Further, we investigated whether the presence of SF is independently associated with device failure in the setting of stent/scaffold restenosis and thrombosis.

### 2.2. Patients

We retrospectively analysed OCT images of all consecutive patients treated at one of three high-volume centres in Germany (University of Mainz; University of Giessen; University of Erlangen) in whom an OCT of one or more stent(s) or BRS(s) was available. OCT was performed for routine (follow-up) control in an elective setting without evidence of ischemia following stent implantation.

A separate group of consecutive patients, in whom OCT was performed in the setting of device failure (restenosis or thrombosis) causing ischemia in the territory downstream were entered in a separate database. Device thrombosis and restenosis were centrally adjudicated based on the analysis of all available clinical data, including coronary angiograms, electrocardiograms and laboratory values. *Academic Research Consortium* criteria were used for thrombosis [17].

Patient and procedural data were entered retrospectively. All data were entered in the multicentric database in an anonymized way according to national privacy policies and laws and following the requirements of the local ethics committees. Data were audited centrally for consistency and plausibility, queries were generated when necessary.

### 2.3. Definitions

*Frequency-domain OCT* was performed using the Ilumien Optis system (St. Jude Medical, Inc., Minneapolis, MN, USA). OCT imaging catheters were placed distally to the segments of interest, and the 54 mm (high resolution) pullback was recorded. If necessary, two sequential pullbacks were acquired, and the segment was reconstructed using conspicuous points (e.g., side branches). All measurements were performed offline using the QCU-CMS software (Medis, Leiden, Netherlands) in Mainz core laboratory using standardized operating procedures and definitions (detailed in [14,18]). Cross sections were analysed at 1 mm longitudinal intervals. The eccentricity index was computed for the lumen and for the device as the ratio between the minimum and the maximum diameters; the symmetry index was defined as the difference between maximum device/lumen diameter and minimum device/lumen diameter divided by the maximum diameter. Qualitative analysis (performed frame by frame) included the presence of evaginations, peri-strut low-intensity areas (PSLIA), calcific nodules, subintimal calcifications (classified as none/arc smaller than 180°/arc larger than 180°), uncovered struts (>6% of the total [19]), malapposed struts and microvessels. Briefly, evaginations were defined as any outwards protrusion in the luminal vessel contour beyond the struts’ abluminal surface between well-apposed struts. Malapposition was defined as a lack of contact of at least one strut with the underlying vessel wall (in the absence of a side branch) [20]. Microvessels were defined as sharply delimitated signal-poor lacunae that extended over multiple contiguous frames. PSLIAs were defined as homogenously appearing, non-signal-attenuating zones around struts of lower intensity than the surrounding tissue. Peri-strut intensity was measured at the mid-strut, based on intensity of the “key” component of the CMYK color model based on raw cross-sectional images.

### 2.4. Strut Fractures

Analysis was performed on all frames. Based on a previous pathology classification, SF was recorded and classified as pattern 1 to 4 (Figure 1):-Pattern 1: one single stacked strut-Pattern 2: two or more stacked struts without deformation-Pattern 3: deformation with evidence of isolated (malapposed) struts or groups of struts not fitting the normal circular geometry of the scaffold in one or more cross sections-Pattern 4: transection with malalignment of the stent segments with or without gap (at least 2 consecutive frames without any strut) [14].

### 2.5. Statistical Analysis

Statistical analysis was performed using IBM SPSS Statistics (SPSS Statistics 23, IBM Deutschland GmbH, Ehningen, Germany) and Medcalc (Mariakerke, Belgium). Categorical data are presented as absolute numbers and percentages. Continuous variables are given as mean (SD) or median (IQR). The frequencies of categorical variables were compared by the Pearson chi-square test, and the distribution of continuous variables was compared by the Mann–Whitney–Wilcoxon test. Univariate and multivariable logistic regression analysis was performed to evaluate the impact of each of the above parameters on the occurrence of SF. The association between SF and device failure was also tested with logistic regression analysis; this analysis was the primary endpoint of the study. The time between implantation and OCT was used as a covariate for all analyses. The threshold for statistical significance was *p* < 0.05.

## 3. Results

### 3.1. Prevalence of SFs in DES after Implantation, during Follow-Up and in the Presence of Device Failure

Patient and procedural characteristics are presented in Table 1.

A total of 185 newer-generation DES (160 patients) were imaged immediately after implantation or as elective 12-month exams. SFs were found in 13 of 159 DES analysed immediately after implantation (8.2%). This prevalence increased to 30.8% (8 of 26) in newer-generation DES analysed at 12 months after implantation (*p* = 0.003, *p* < 0.0001).

### 3.2. Patterns of SF and Association with OCT Characteristics

The patterns of SF were evenly distributed (Figure 2) without differences between immediate controls and follow-up controls. In five cases (24%), multiple fracture patterns were present in the same device. Devices with SF were more often implanted in bifurcation lesions (*p* = 0.024), and stents and vessels tended to be smaller (*p* = 0.003). Features of device asymmetry and eccentricity were associated with SF (Table 1, Appendix A, Figure 3 and Appendix A). Fractures were associated (Table 1 and Figure 4) with the presence of microvessels (*p* < 0.001), uncovered struts (*p* < 0.001), evaginations (*p* = 0.001), peri-strut low-intensity areas (*p* = 0.001) and subintimal calcium (*p* < 0.001). There was no difference across different fracture patterns in the prevalence of these findings (Appendix A). Postdilatation at index was performed more frequently in fracture pattern 1 (55% vs. 12% in pattern 4, *p* = 0.01), and evaginations were more commonly associated with pattern 4 (55%) than pattern 1 (11%, *p* = 0.03). There was no other difference across patterns.

### 3.3. Procedural Parameters Associated with SF

The results of the uni- and multivariate logistic regression analysis are shown in Appendix A. Stenting in bifurcation and calcific lesions (OR: 3.5 [1.1–11.0], *p* = 0.03 and OR: 4.0 [2.2–7.2], *p* < 0.001) was independently associated with SF.

### 3.4. Prevalence, Characteristics and Association with Procedural Parameters in BRS

Briefly, a total of 289 BRS (242 patients) were imaged post-implantation (*n* = 192) or as elective follow-up exam (*n* = 97). SF was diagnosed in 75 (30.0%) of these devices (65 patients). The prevalence of BRS SF was 8.5% immediately after implantation (13 BRS of 153) and raised to 42.5% (62 of 146) at 12-month follow-up (*p* < 0.0001).

### 3.5. The Impact of Fractures on Device Failure

A total of 56 cases (32 DES, 24 BRS) of device failure were analysed. We found that 36 devices showed evidence of restenosis, 20 of thrombosis. The characteristics of these patients are presented in Appendix A (description of individual cases). The prevalence of SF was 61.2% in DES and 66.7% in BRS (*p* < 0.0001 vs. controls), without difference between restenosis and thromboses. In contrast to control devices, the prevalence of pattern 4 (gap) increased by 7.5 times in DES and by 12.2 times in BRS (*p* < 0.0001). Also, the prevalence of SF was not different between DES and BRS (*p* = 0.625). Figure 4 presents the prevalence of pathologic findings at OCT, and Figure 5 presents examples describing the spatial relationship between SF and device failure. In all cases, SFs were found in the segment presenting the pathological finding. In multivariate logistic analysis, SFs were associated with DES failure, with an OR of 12.5(95%CI 5.3–29.4, *p* < 0.0001, Appendix A). A similar association was shown for BRS (OR: 7.2 [2.2–23.2, *p* = 0.001).

## 4. Discussion

Stent failure remains an issue for many patients treated with coronary stents. The pathophysiology of this type of complication is complex, as it is influenced by patient-related, procedure-related and device-related factors [21,22,23]. Given the steadily increasing number of interventions performed worldwide and the severity of some of these complications, it is important to further investigate their possible mechanisms. We investigated the prevalence and associations of strut fractures detected by OCT. The major findings of this paper include:(1)incidental findings of fractures occurred in ~8% of new-generation drug-eluting stents immediately after implantation, and this rate was as high as ~60% in the setting of device failure;(2)parameters of lesion/vessel anatomy, including bifurcation and calcific lesions were associated with fracture; increased asymmetry and eccentricity were associated with SF;(3)we propose an OCT classification based on a previously published pathological staging which allows distinguishing different degrees of SF. Using this classification, we found that the prevalence of pattern 4 (gap) SF increased by ~10 times in device failure compared to control devices;(4)fractures were associated with other OCT abnormalities, including peri-strut low-intensity areas, uncovered and/or malapposed struts (all suggestive of incomplete stent healing);(5)the presence of fractures was independently associated with device failure;(6)similar results were observed in BRS. Of note, SFs represent a step of the bioresorption of scaffolds, while they are unwanted phenomena in stents. However, the fact that SFs were associated with PSLIA, uncovered/malapposed struts and ultimately device failure in both device types supports the concept that SFs represent a risk factor for device failure in both settings.

### 4.1. Evidence on SF

Anecdotal evidence has long reported the existence of an association between SFs and device failure, including restenosis and thrombosis [8,24]. The true incidence of this phenomenon and its mechanism(s) remain however underexplored. Previous studies have reported markedly heterogeneous rates, varying between 1% and ~20% based on the clinical setting and definitions used [3,4,5,6,7,9,15]. Compatible with our data, stent fractures were observed in 29% of lesions in a pathology study, which is far more frequent than that generally reported clinically. This difference reflects likely the higher resolution of OCT and pathology compared to angiography. The prevalence of adverse pathological findings such as inflammation, neointima and thrombosis was higher in the presence of SFs, without significant differences across fracture patterns [16]. The authors note that this observation might depend on a referral bias inherent to the fact that the incidence of these findings would be expected to be higher in a pathology series. Our finding of an association between SF and acute coronary syndrome, calcifications and implantation in the RCA (right coronary artery) is in line with previous reports [4,25,26,27,28,29,30]. Adding to these notions, we report an association of SF with device/vessel eccentricity and asymmetry, compatible with an increased mechanical stress on these devices as a cause of SF. Finally, strut discontinuations (analogous to our pattern 3) were reported in the INVEST registry [31].

Based on the pathology study mentioned above, we propose a classification of the SF patterns. In our series, more complex SF patterns were relatively more common in the presence of device failure, but there was no evidence of an impact on the other OCT characteristics. Although the mechanism of the association between SF and adverse events remains unclear, inflammatory reactions associated with the mechanical stress and/or disturbances in blood flow dynamics have been proposed as possible mediators.

### 4.2. Limitations

This is a hypothesis-generating study with a small sample size. Given its non-interventional design, it is impossible to infer causality, and only association can be described. Further, early and (even more) late events that follow stent implantation have a complex pathophysiology and often cannot be reduced to one single mechanism. The current study investigated the role of SFs, but the importance of other factors (patient characteristics, procedural parameters, therapy, etc.) cannot be emphasized enough. Since this retrospective study included all consecutive patients with an OCT, but not all patients undergoing percutaneous coronary intervention during the same time period, the risk of a referral bias needs to be acknowledged. Furthermore, the role of vessel motion and vessel tortuosity cannot be assessed using OCT [27]. The time from implantation to assessment of SF has been traditionally reported to be a predictor of SF. Since our study was based on a single cross-sectional assessment, we cannot draw conclusions on the incidence of SF over time, particularly in DES, given the small number of controls a long term after implant. We included time from implantation as a co-variate for all analyses. However, the cross-sectional nature of the study does not allow a statement on the true timing of the SF. Finally, a number of mechanisms of stent/scaffold thrombosis is known [31,32], and we do not want to imply that SFs are the only (or prevalent) one.

## 5. Conclusions

In this hypothesis-generating, multicentric series of consecutive patients, the incidence of OCT-diagnosed SF was ~20% in control devices, suggesting that this is a common phenomenon which often has no implications. This prevalence, however, increased by over three-fold in the setting of device failure. Predictors of SF were implantation in an acute setting, in the RCA, and in calcific vessels. Fractures were associated with evidence of incomplete healing and device complications in both stents and scaffolds and showed a strong association with restenosis and thrombosis.

## Figures and Tables

**Figure 1 jcm-10-01765-f001:**
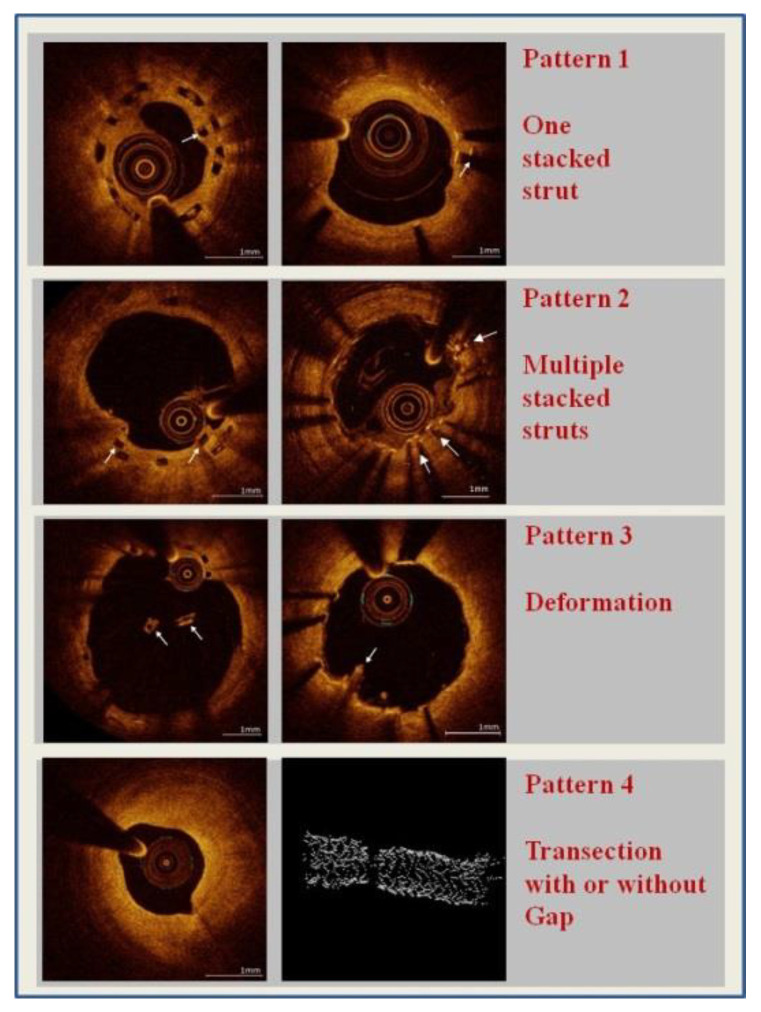
The different patterns of strut fracture (SF) in 2D and (in the case of pattern 4) 3D optical coherence tomography (OCT) imaging.

**Figure 2 jcm-10-01765-f002:**
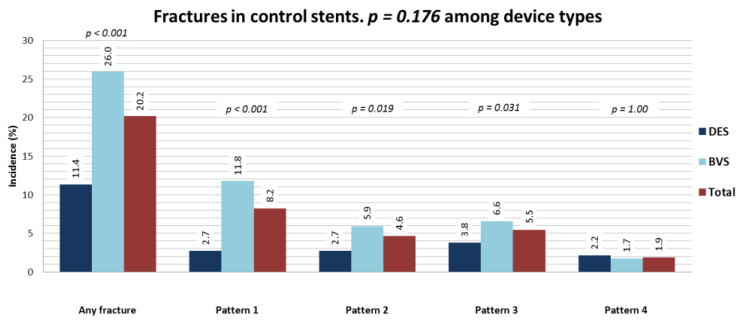
Prevalence of SF, distribution across patterns and differences between stents and scaffolds.

**Figure 3 jcm-10-01765-f003:**
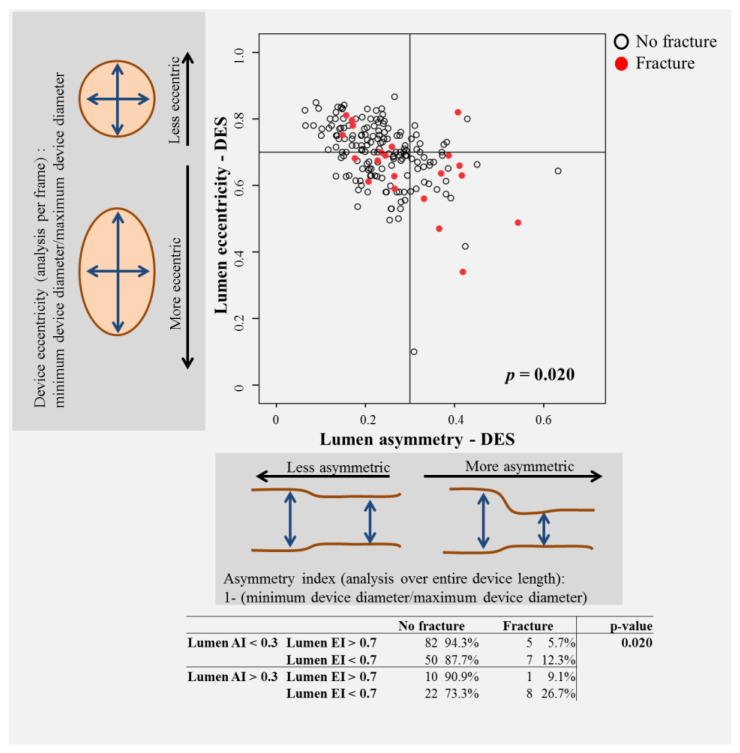
The impact of asymmetry and eccentricity on the prevalence of SF in DES. Both parameters had an impact, and the combination of both had an additive impact.

**Figure 4 jcm-10-01765-f004:**
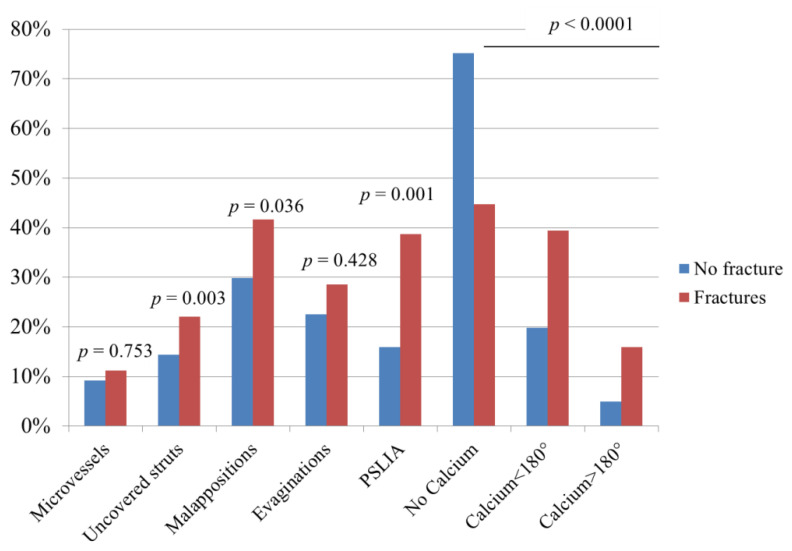
Prevalence of pathological findings in the presence of SF.

**Figure 5 jcm-10-01765-f005:**
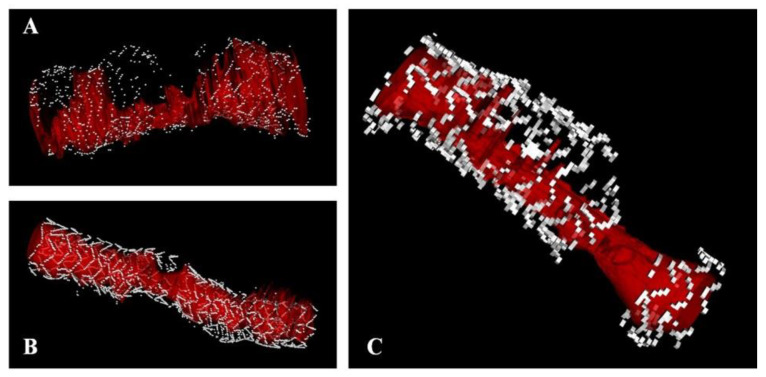
Three examples of the spatial relationship between restenosis/thrombosis and strut fracture. The figure presents three cases of pattern 4 ((**A**,**B**): DES; (**C**): BRS) fracture associated with device failure ((**A**): thrombosis; (**B**,**C**): restenosis). The residual vessel lumen is presented in red.

**Table 1 jcm-10-01765-t001:** Patient characteristics.

DES-Control Patients Only					
	No Fracture (*n* = 141)		Fracture (*n* = 19)		*p*-Value
Clinical Presentation (per patient)	mean/median/*n*	SD/IQR/%	mean/median/*n*	SD/IQR/%	
Male	105	74.5%	14	73.7%	0.941
Age (years)	66.0	17.0	62.0	24.0	0.565
Positive family history	48	34.0%	6	31.6%	0.831
Diabetes mellitus	47	33.3%	7	36.8%	0.761
Hypertension	109	77.3%	14	73.7%	0.725
Hyperlipidemia	76	53.9%	8	42.1%	0.334
Active smoker	42	29.8%	10	52.6%	0.129
Indication for stent implantation					0.399
Planned	21	15.9%	2	10.5%	
Stable angina	56	42.4%	9	47.4%	
Instable angina	15	11.4%	0	0.0%	
NSTEMI	19	14.4%	5	26.3%	
STEMI	21	15.9%	3	15.8%	
LVEF	55	10	55	5	0.903
*n*_vessel-disease					0.066
1-vessel	43	30.7%	1	5.3%	
2-vessel	55	39.3%	10	52.6%	
3-vessel	42	30.0%	8	42.1%	
Antiplatelet Therapy					0.148
Clopidogrel	80	58%	7	38%	
Prasugrel	29	21%	5	28%	
Ticagrelor	30	22%	6	33%	
**Angiographic Characteristics (per device)**	***n* = 164**		***n* = 21**		
treated vessel					0.905
RCA	52	31.7%	8	38.1%	
LAD	84	51.2%	9	42.9%	
LCX	22	13.4%	3	14.3%	
LM	6	3.7%	1	4.8%	
ACC/AHA classification					0.853
Type A	23	15.2%	2	12.5%	
Type B1	34	22.5%	5	31.3%	
Type B2	67	44.4%	7	43.8%	
Type C	27	17.9%	2	12.5%	
De novo lesion	125	76.2%	3	14.3%	0.328
Implantation on thrombus	18	11.0%	2	9.5%	0.840
Implantation in CTO	4	2.4%	0	0.0%	0.469
Implantation with overlap	42	25.6%	6	28.6%	0.771
Implantation on bifurcation	18	11.0%	6	28.6%	0.024
Predilatation	119	72.6%	18	85.7%	0.195
Ballon diameter (mm)	2.8	0.5	2.8	0.3	0.167
Ballon length (mm)	15.0	8.0	15.0	12.0	0.989
Predilatation pressure (atm)	14.6	3.5	12.8	4.4	0.418
Diameter stent	3.2	0.5	2.9	0.2	0.003
Length stent (mm)	18.0	13.0	28.0	15.0	0.429
Implantation pressure (atm)	13.8	2.3	12.8	1.8	0.248
Postdilatation	98	59.8%	9	42.9%	0.140
Ballon diameter (mm)	3.3	0.6	3.1	0.4	0.311
Ballon length (mm)	14.0	3.0	12.0	6.0	0.824
Postdilatation pressure (atm)	16.1	5.2	13.6	3.8	0.109
OCT analysis					
avg *n*_struts/frame	9.6	2.5	9.0	3.2	0.460
Length OCT (mm)	18.8	12.5	28.2	6.4	0.490
**Maximal Area (mm^2^)**					
Lumen	10.5	3.4	8.4	1.8	0.449
Vessel	10.6	3.0	9.0	1.7	0.992
Stent	9.7	2.9	8.3	1.7	0.949
**Maximal Diameter (mm)**					
Lumen	3.6	0.6	3.3	0.3	0.459
Vessel	3.6	0.5	3.4	0.3	0.998
Stent	3.5	0.5	3.2	0.3	0.958
**Minimal Area (mm^2^)**					
Lumen	5.9	2.0	4.4	1.4	0.017
Vessel	6.5	2.1	5.0	1.6	0.082
Stent	5.9	2.0	4.4	1.5	0.079
**Minimal Diameter (mm)**					
Lumen	2.7	0.5	2.3	0.4	0.013
Vessel	2.9	0.5	2.5	0.4	0.076
Stent	2.7	0.5	2.3	0.4	0.073
**Avg Area (mm^2^)**					
Lumen	7.7	2.2	6.2	1.5	0.274
Vessel	8.4	2.3	7.0	1.3	0.492
Stent	7.6	2.2	6.3	1.3	0.502
Neointima	−0.1	0.5	−0.08	0.8	0.010
**Avg Diameter (mm)**					
Lumen	3.1	0.5	2.8	0.3	0.259
Vessel	3.2	0.4	3.0	0.3	0.518
Stent	3.1	0.4	2.8	0.3	0.528
**Avg Stenosis (%)**					
Stenosis area	8.7	6.0	7.5	11.8	0.018
Stenosis diameter	4.5	2.9	3.8	6.4	0.018
Device/artery ratio	1.05	0.12	1.11	0.18	0.071
**Eccentricity and Asymmetry**					
Lumen AI > 0.3	32	19.5%	9	42.9%	0.015
Stent AI > 0.3	19	11.6%	6	28.6%	0.032
Lumen EI < 0.7	72	43.9%	15	71.4%	0.017
Stent EI < 0.7	35	21.3%	8	38.1%	0.087
Maximal lumen asymmetry	0.25	0.13	0.26	0.07	0.027
Maximal stent asymmetry	0.22	0.12	0.26	0.11	0.009
Maximal lumen eccentricity	0.70	0.12	0.69	0.09	0.089
Maximal stente ccentricity	0.75	0.12	0.74	0.15	0.322
**Qualitative Analysis**					
Microvessels	2	6.9%	2	18.2%	<0.001
Uncovered struts	2	6.9%	7	63.6%	<0.001
Malappositions	66	40.5%	11	52.4%	0.545
Evaginations	8	26.7%	2	18.2%	0.001
PSLIA	6	20.7%	5	41.7%	0.001
No Calcium	113	73.9%	4	20.0%	<0.001
Calcium < 180°	32	20.9%	9	45.0%	
Calcium > 180°	8	5.2%	7	35.0%	

DES, drug-eluting stents, NSTEMI: non-ST elevation myocardial infarction, LVEF: left ventricular ejection fraction, RCA: right coronary artery, RIVA: left anterior descending, LCX: left circumflex, LM: left main, ACC/AHA: american college/american heart association, AI: asymmetry index, EI: eccentricity index, PSLIA: peri-strut low-intensity areas; CTO: chronic total occlusion; OCT: optical coherence tomography.

## Data Availability

Data will be made available upon justified request.

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
