# Peer review of "Coronary Stent Strut Fractures: Classification, Prevalence and Clinical Associations"

_jcm, 2021, doi:10.3390/jcm10081765_

Round 1
Reviewer 1 Report
Coronary stent strut fractures: Classification, Prevalence and clinical associations.
Dear Editor, thank you for the opportunity to review this article.
The authors do excellent multi-center retrospective work. The purpose of the study is to describe the incidence of drug eluting stent post fractures and the coronary lesions and / or features with which they were most frequently associated. This kind of synthesis work is very useful. It is very well written and structured.
Author Response
We thank very much for your positive comments.
Reviewer 2 Report
Schochlow et al. performed elegant work on the impact of stent fracture on stent outcome.
I have some minor considerations:
1) Add in the table the type of antiplatelet therapy performed
2) It is not clear from table 1 whether the diagnosis of presentation is related to a problem with the stent being evaluated or it is events on other vessels, please specify
Author Response
Dear Reviewer,
thank you very much for your comments. To your points:
- The P2Y12 used has been added. Duration of therapy was recommended based on the guidelines. There was no difference between fracture and non-fracture (P=0.148)
2. You are right. This has been changed to "indication for stent implantation". Also, "planned" has been chaged to "staged procedure". These were patients who received the stent under study as a staged procedure after treatment of another lesion in the setting of an acute coronary syndrome.
We hope you will find these changes satisfactory.
Reviewer 3 Report
In this study, K. Schochlow et al. report a prevalence of OCT-diagnosed stent fractures unexpectedly high (20%) in the setting of elective controls, and increased by ~3-fold in the setting of device failure such as restenosis and thrombosis. Despite the intrinsic limitations of this observational cross-sectional study (small number of patients and no analysis of stent fracture incidence over time) , we acknowledge that these findings obtained by OCT reconstruction of stented coronary segment, allowing detailed imaging of fractures by severity grade, are relevant for a better understanding of device failure, bioreabsorbable scaffolds included, a main issue in CAD treatment (see J Am Coll Cardiol 2020;75:590–604, ). However, an improved, more extensive discussion of fractures-related vs fractures-unrelated adverse events in stented CAD, with special regard to very late events (stent-related events occur in 5.0% of patients after newer generation stenting within the first year and in 7.7% of patients between years 1 and 5) would be advisable (the authors themselves found fractures in 8.2% DES analysed immediately after implantation and in 30.8% of newer-generation DES analysed at 12 months after implantation) . Indeed, in real clinical world, with a raising number of electively stented CAD patients, long-term outcome of implanted devices is a major concern : better monitoring of stent integrity for timely restenting and/or appropriate pharmacological treatment is a key point in management and risk stratification of the so called "stent disease" , a widespread clinical issue and a major obstacle to a "preventive plaque sealing" approach to MACE (J Am Coll Cardiol Intv 2021;14:461–7).
Minor comment
typo on line 201 "OR: 72 [2.2-23.2]
Author Response
Dear reviewer 3,
we are grateful for your comments.
To your points:
- we acknowledge that these findings obtained by OCT reconstruction of stented coronary segment, allowing detailed imaging of fractures by severity grade, are relevant for a better understanding of device failure [...] However, an improved, more extensive discussion of fractures-related vs fractures-unrelated adverse events in stented CAD [...] would be advisable[...]
The reviewer is absolutely correct. We have added a sentence to this regard in the limitations section:
Early and (even more) late events that follow stent implantation have a complex pathophysiology and often cannot be reduced to one single mechanism. The current study investigates the role of SFs, but the importance of other factors (patient characteristics, procedural parameters, therapy etc) cannot be emphasized enough.
and in the discussion:
Stent failure remains an issue for many patients treated with coronary stents. The pathophysiology of this type of complications is complex as it is influenced by patient-related, procedure-related, and device-related factors[21-23]. Given the steadily increasing number of interventions performed worldwide and the severity of some of these complications, it is important to further investigate their possible mechanisms.
We do not in any way reduce the mechanism of stent failure to one simple factor, stent fracture, but we do want to describe an association which we observe in this database. At the same time, given the length of the manuscript and the complexity of the topic, we do have to admit that much larger studies are necessary to investigate the interactions among all factors.
2. Minor comment
typo on line 201 "OR: 72 [2.2-23.2]
This has been corrected, thank you.
This manuscript is a resubmission of an earlier submission. The following is a list of the peer review reports and author responses from that submission.